# Macroscale intrinsic dynamics are associated with microcircuit function in focal and generalized epilepsies

Siqi Yang[1,2,3 ✉], Yimin Zhou[1], Chengzong Peng[1], Yao Meng[2], Huafu Chen [2], Shaoshi Zhang[3], Xiaolu Kong[3], Ru Kong[3], B. T. Thomas Yeo[3], Wei Liao [2 ✉] & Zhiqiang Zhang [4 ✉]

Epilepsies are a group of neurological disorders characterized by abnormal spontaneous brain activity, involving multiscale changes in brain functional organizations. However, it is not clear to what extent the epilepsy-related perturbations of spontaneous brain activity affect macroscale intrinsic dynamics and microcircuit organizations, that supports their pathological relevance. We collect a sample of patients with temporal lobe epilepsy (TLE) and genetic generalized epilepsy with tonic-clonic seizure (GTCS), as well as healthy controls. We extract massive temporal features of fMRI BOLD time-series to characterize macroscale intrinsic dynamics, and simulate microcircuit neuronal dynamics used a large-scale biological model. Here we show whether macroscale intrinsic dynamics and microcircuit dysfunction are differed in epilepsies, and how these changes are linked. Differences in macroscale gradient of time-series features are prominent in the primary network and default mode network in TLE and GTCS. Biophysical simulations indicate reduced recurrent connection within somato-motor microcircuits in both subtypes, and even more reduced in GTCS. We further demonstrate strong spatial correlations between differences in the gradient of macroscale intrinsic dynamics and microcircuit dysfunction in epilepsies. These results emphasize the impact of abnormal neuronal activity on primary network and high-order networks, suggesting a systematic abnormality of brain hierarchical organization.

[1] School of Cybersecurity (Xin Gu Industrial College), Chengdu University of Information Technology, Chengdu 610225, PR China. [2] The Clinical Hospital of Chengdu Brain Science Institute, School of Life Science and Technology, University of Electronic Science and Technology of China, Chengdu 610054, PR China. [3] Centre for Sleep and Cognition (CSC) & Centre for Translational Magnetic Resonance Research (TMR), Yong Loo Lin School of Medicine, National University of Singapore, Singapore, Singapore. [4] Laboratory of Neuroimaging, Department of Radiology, Jinling Hospital, Nanjing University School of Medicine, Nanjing 210002, PR China. ✉email: fmriyangsq@163.com; weiliao.wl@gmail.com; zhangzq2001@126.com

The brain is a complex connectome that is organized hierarchically. Cortical areas are anatomically connected and synaptically interacted with each other by neuronal populations[1]. Converging evidence suggests that multiscale brain organizations follow sensory-association gradient, including gene expression[2], cortical myelin[3], and layer characteristics[4], as well as functional connectivity[5]. Nevertheless, the impact of brain dysfunction caused by abnormal neuronal activity is not fully understood. How this dysfunction affects the spatial arrangement of cortical gradient and macroscale spontaneous brain activity remains unknown.

Epilepsy is a powerful model for studying the influence of neuronal activities on the spontaneous brain function. The System Epilepsies hypothesis[6] proposes that the enduring susceptibility to generate seizures[7] in some epilepsies is due to the specific vulnerability of a system as a whole, emphasizing the importance to understanding the pathology from a systematic perspective. Abnormal firing of neurons causes local circuit dysfunction, leading to alterations of macroscale functional activities during seizure propagation[8]. The Virtual Brain model has demonstrated that a combination of a global shift in the brain's dynamic equilibrium and locally hyperexcitable network nodes provides a mechanistic explanation for the epileptic brain during interictal resting state[9]. The spontaneous brain activity during the resting state provides a specific and multifaceted fingerprint of brain function[10], as different metrics represent multi-dimensional features of neuronal activity. Altered static and dynamic amplitude of low-frequency fluctuations reflect the intensity of spontaneous fluctuations in BOLD functional magnetic imaging (fMRI) in epileptic brains[11,12], and are related to interictal epileptiform discharges[13]. Abnormal regional homogeneity reveals functional integration of epileptic lesions with neighboring regions[14], and further network analyses extend the impact of local lesions to widespread distributed changes[15–17]. The network communication analysis revealed that the enhanced structure-function correlation may be related to the smaller functional repertoire in TLE, while sparing the central core of the brain, which may represent a pathway that promotes the spread of seizures[18]. One limitation is that these conventional functional metrics are based on specific and selected features of time-series. Therefore, multi-dimensional features of time-series can help comprehensively characterize spontaneous brain activity in epileptic brain.

Large-scale biophysical models of coupled brain regions can bridge microscale and macroscale brain organizations[19–21]. These models describe the collective action of neuronal populations, and have been successful in modeling brain activity during seizures[22], sleep[23], and the resting-state[24,25]. Previous studies on epileptic brain using the relaxed mean field model found increased external input in generalized epilepsy, and decreased external input and increased recurrent connection in temporal lobe epilepsy, suggesting a dissociation between focal and generalized epilepsies derived by different microcircuit function[26]. However, the relaxed mean field model[27] has not been shown to recapitulate functional connectivity dynamics. The parametric mean field model (pMFM) can simulate more realistic static and dynamic functional connectivity by parameterizing local neuronal properties with anatomical and functional gradients[28]. This model could help to understand how brain structure shapes intrinsic brain dynamics in epilepsy.

In the current study, we included two epileptic subtypes: patients with temporal lobe epilepsy (TLE, $n = 75$) and patients with genetic generalized epilepsy with tonic-clonic seizures (GTCS, $n = 79$), as well as sex-, age-matched healthy controls (HC, $n = 108$). We extracted massive features of time-series to comprehensively describe the spontaneous brain activity. We then performed principal component analysis to capture the macroscale cortical gradient of brain intrinsic dynamics. In addition, we used a parametric mean field model to simulate microcircuit function, including recurrent connections and external input. We then analyzed the association between these multiscale brain function in epilepsies.

## Results

**Macroscale intrinsic dynamics and microcircuit function.** To describe the macroscale intrinsic dynamics, we first comprehensively described summary features of intrinsic BOLD fMRI signals across the cerebral cortex. We used the Desikan–Killiany anatomical parcellation[29] with 68 cortical regions to extract time-series. Using highly comparative time-series analysis, the hctsa toolbox[30], we extracted massive temporal features, yielding over 7,000 features in each regional time-series. To investigate the macroscale topographic organization of time-series features, we sought to identify the embedding axis that explained maximal variance. We therefore performed principal component analysis, capturing spatial gradients of shared time-series features matrix (5,360 × 68) across three groups. From the perspective of dimensionality reduction method, gradient more focused on the relationship between brain regions regard to their similarity of 5,000 shared temporal features, which representing the fingerprints of regional intrinsic dynamics. We therefore emphasized the gradient of temporal features rather than temporal correlation of time-series. These gradients reflected macroscale intrinsic dynamics across the cerebral cortex (Fig. 1a).

To estimate the microcircuit function across the cerebral cortex, we used the parameter mean field model (pMFM)[28] to simulate local synaptic properties, the recurrent connectivity (RC) and the external input (I) (Fig. 1b). We computed the group-averaged functional connectivity (FC) and structural connectivity with Desikan–Killiany atlas in each set, and computed functional connectivity dynamics (FCD) for each participant. In the pMFM, the neural dynamics of each brain region are driven by four components, (i) intra-regional input, (2) external input, (3) inter-regional input, and (4) neuronal noise. In the model, the parameters RC, I, and σ are set as a linear combination of the FC gradient, resulting 7 unknown coefficients. The 7 unknown coefficients were estimated by minimizing disagreement (1 – Pearson's correlation r + Kolmogorov-Smirnov) between the empirical and simulated FC and FCD matrices (details see Methods).

**Macroscale intrinsic dynamics in epilepsies.** Macroscale intrinsic dynamics were reflected by the gradients of time-series features. The spatial distribution of the top two components (PC1 and PC2) in epilepsies and control group is shown in Fig. 2a and b, and the variance explained by the sorted first ten components is shown in Fig. 2c. The PC1 after Procrustes alignment shows a ventromedial-lateral gradient, and the PC2 captures the sensory-association gradient. To directly observe the perturbation in embedding space in epilepsies, we charted the three groups in a coordinate system spanned by the top two gradients (Fig. 2d). We found different perturbations in TLE and GTCS compared to HC, reflecting functional dynamics heterogeneity in two epileptic subtypes.

We then localized parcel-wise and network-wise alterations of macroscale intrinsic dynamics in TLE and GTCS. We performed group comparisons in gradients between epilepsies with HC with 5,000 permutations ($P < 0.05$, family-wise error correction), after Procrustes alignment with control. For the PC1, both TLE and GTCS groups showed similar spatial alterations, with increased intrinsic dynamics in somatomotor and visual cortices, and decreased in cingulate cortices. For the PC2, the two epileptic

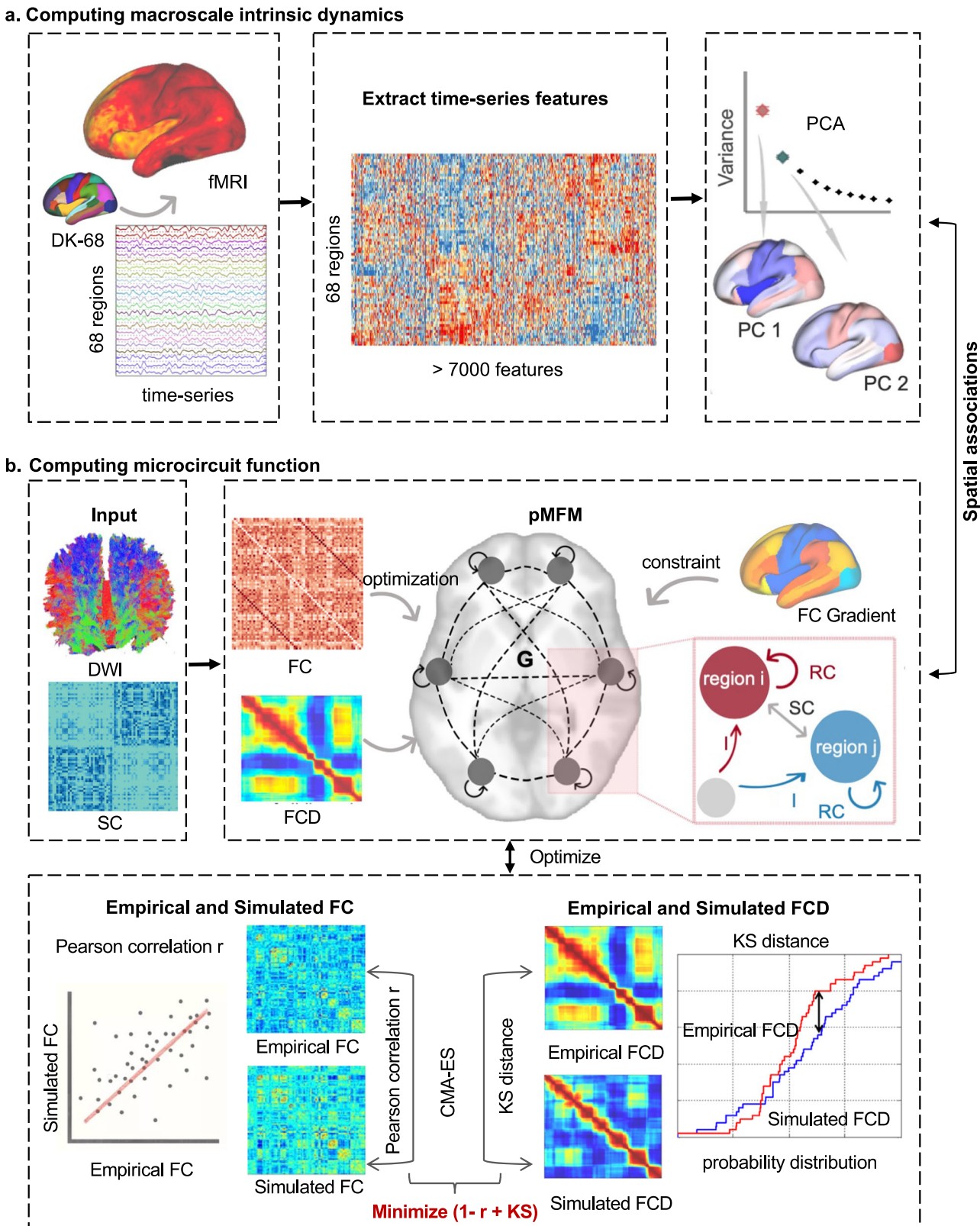

**Fig. 1 Computing macroscale intrinsic dynamics and microcircuit function. a** Using the Desikan–Killiany atlas to extract the fMRI BOLD across the cerebral cortex, then performing the highly comparative time-series analysis (*hctsa*) toolbox to extract massive time-series features, and applying principal component analysis to capture the spatial gradients of time-series features. **b** In the pMFM, group-averaged structural connectivity was the input, and parameters RC, I, and σ were set as a linear combination of the FC gradient. The covariance matrix adaption evolution strategy algorithm was applied to estimate the model by minimizing the minimizing disagreement between the empirical and simulated FC and FCD matrices. The agreement between empirical and simulated FC was maximize the Pearson correlation r, the agreement between empirical and simulated FCD was minimize the Kolmogorov-Smirnov distance between the upper triangular entries of the two FCD matrices.

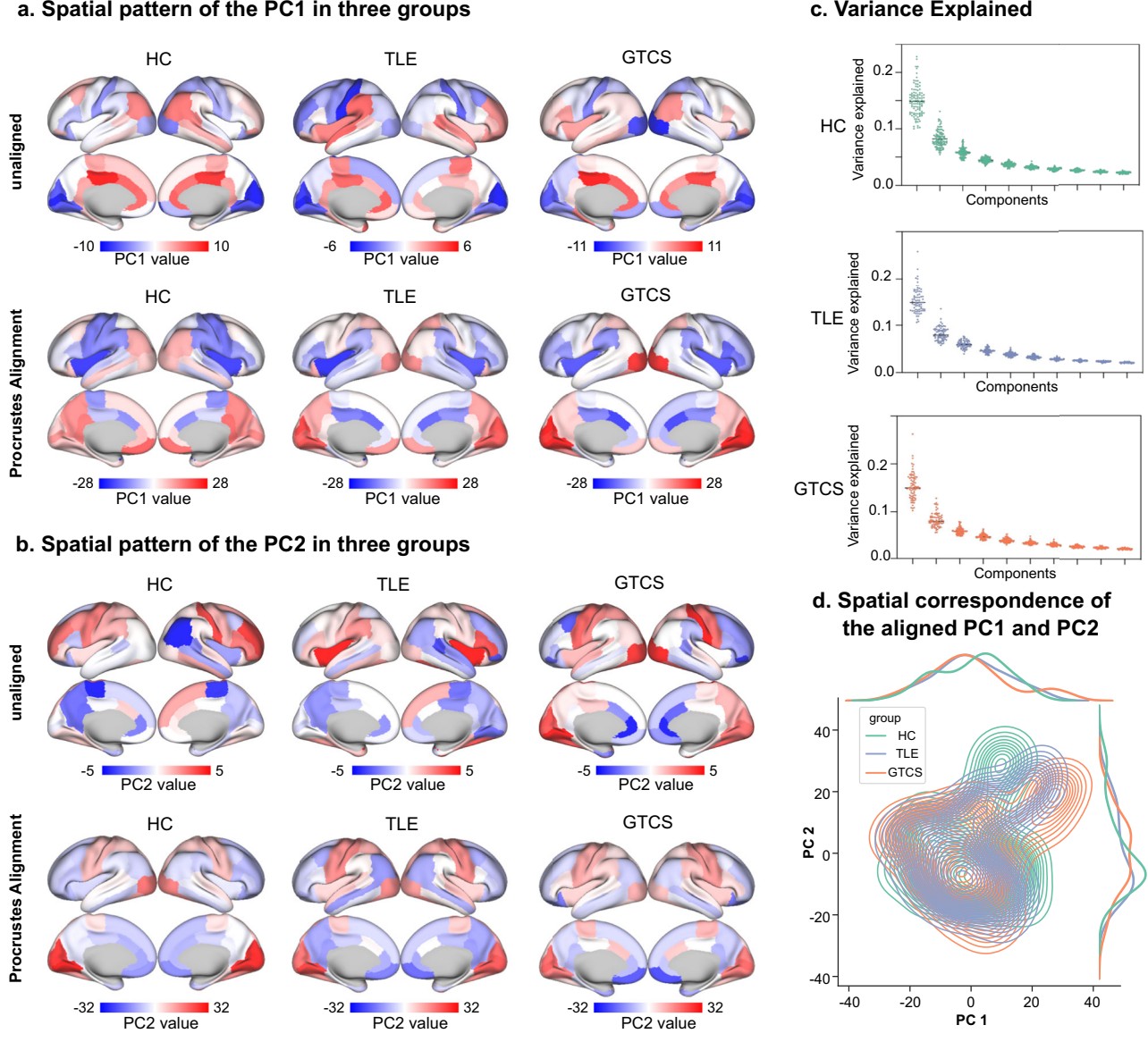

**Fig. 2 Macroscale intrinsic dynamics in three groups.** Spatial pattern of the PC1 (**a**) and PC2 (**b**) in each group before and after Procrustes alignment. **c** Explained variance of the sorted components. **d** Three groups in a coordinate system spanned by the top two gradients. HC healthy controls, TLE temporal lobe epilepsy, GTCS genetic generalized epilepsy with tonic-clonic seizures, PC1 first component, PC2 second component.

subtypes showed different spatial alterations. In TLE, increased intrinsic dynamics were observed in bilateral insula and anterior cingulate cortex, while decreased intrinsic dynamics were observed in left medial temporal gyrus and right precuneus. In GTCS, increased intrinsic dynamics were observed in bilateral insula, supramarginal gyrus, anterior cingulate cortex and superior frontal cortex, while decreased intrinsic dynamics were observed in visual cortex (Fig. 3a).

Next, we sought to determine which features contributed most to these alterations in intrinsic dynamics. To do this, we assessed the spatial correlation between each feature (named "loading" in the hctsa toolbox) and statistical comparison map of epileptic subtypes with the HC group. Each feature described a specific temporal property of a given time-series, including distributional properties, entropy and variability, autocorrelation, and nonlinear properties. The top contributing features were selected based on the highest Pearson correlation and were visualized in word clouds. For PC1, the top contributing features described the stability of time-series in TLE and described the model prediction

of time-series in GTCS. For PC2, the contributing features were sensitive to Fourier frequency characteristics and the stability of time-series in TLE, and sensitive to model predicted stability in GTCS (Fig. 3a).

To localize network-wise alterations in TLE and GTCS, we used two parcellations to compare the macroscale intrinsic dynamics between epilepsies and HC. First, in comparison with the functional community atlas[31], increased PC1 was shown in the primary network in TLE, while decreased PC1 was found in default mode network. In GTCS, multi-network alterations were found. Increased PC2 was shown in the somatomotor network and ventral attention network in TLE, while decreased PC2 was found in the default mode network. In GTCS, increased PC2 was shown in the limbic network and dorsal attention network, while decreased PC2 was found in the visual network (Fig. 3b).

**Microcircuit simulations by the pMFM.** We divided the HC, TLE and GTCS into training, validation, and test sets to optimize

## a. Parcel-wise differences between epileptic subtypes and controls

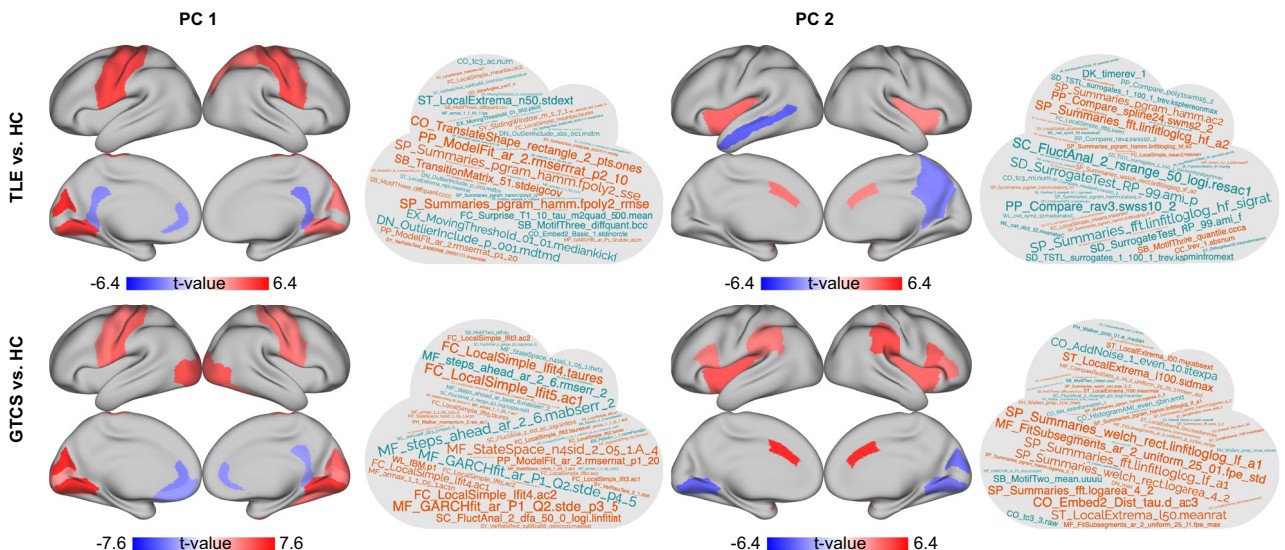

## b. Network-wise differences between epileptic subtypes and controls

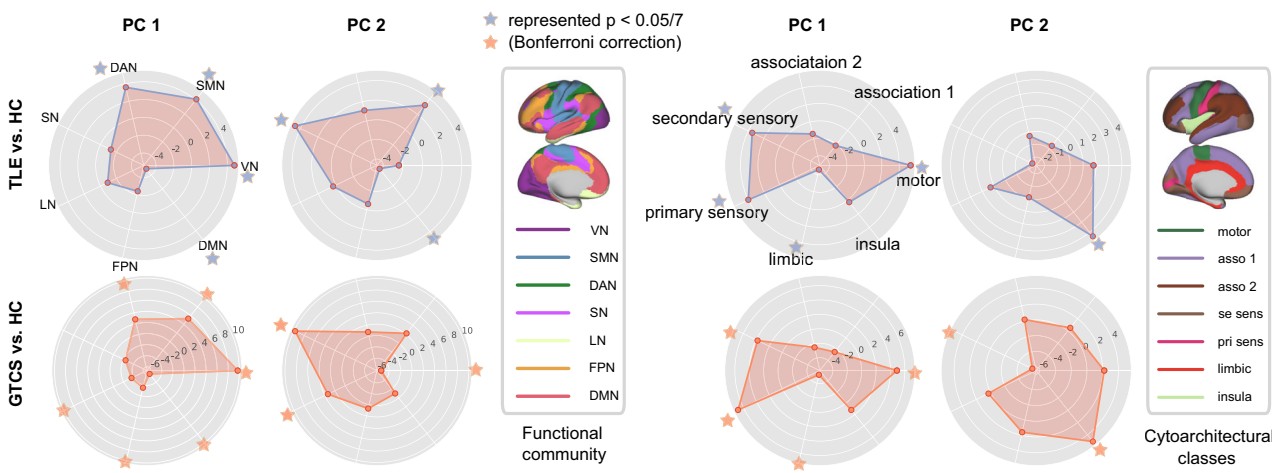

**Fig. 3 Parcel-wise and network-wise comparison of aligned PC1 and PC2 between epilepsies and HC. a** Parcel-wise differences in PC1 and PC2 between TLE/GTCS relative to HC. Red and blue regions indicate significant macroscale intrinsic dynamics in epilepsies relative to controls. The cloud maps indicate top features that contributed most to the alterations in intrinsic dynamics (PC1 and PC2). Orange color represents positive contribution, and green color represents negative contribution. **b** Network-wise differences in PC1 and PC2 were compared using two parcellations, the functional community atlas and the cytoarchitectural atlas. A radar plot shows the difference in intrinsic dynamics corresponding to the functional community/cytoarchitectural class between the epilepsies and HC. The star represents significant difference between groups, $P < 0.05/7$ with Bonferroni correction. VN visual network, SMN somato-motor network, DAN dorsal attention network, VAN ventral attention network, LN limbic network, FPN fronto-parietal network, DMN default mode network.

the pMFM procedure. Demographic and clinical characteristics were matched in each set (Supplementary Table 1). Following the optimization of the pMFM, the RC and I were estimated in each group (Fig. 4a, b). Consistent with previous study[28], recurrent connection gradually varies from sensory-motor to association cortices, and external input reveals a reverse changing with association cortices to sensory-motor regions. After group comparisons, we examined the changes in RC and I between the epilepsies and controls. We observed similar pattern of changes in RC in TLE and GTCS, with lower RC in sensory-motor regions and higher RC in association cortices. However, a different pattern of changes in I were observed in TLE and GTCS, with TLE group showing higher I in sensory-motor regions and lower I in association cortices, and GTCS group showing higher I across the cerebral cortex (Fig. 4c, d). To determine the regions with

significantly statistical changes, we performed 1,000 permutations ($P < 0.05$, false discovery rate correction) by randomly shuffling the epilepsies and controls within training, validation and test set, and then repeated the analysis in the pMFM procedure. We observed significant decreased RC in somatomotor region and fusiform gyrus in TLE, and more widely decreased RC in GTCS (Supplementary Fig. 1).

**Associations in multiscale brain function**. To further examine the associations between the macroscale intrinsic dynamics and microcircuit function in epilepsies, we assessed the spatial correlation between the altered gradients of time-series features and changes of the recurrent connections and external input yielded from a neural dynamics model. Notably, we focused on the spatial

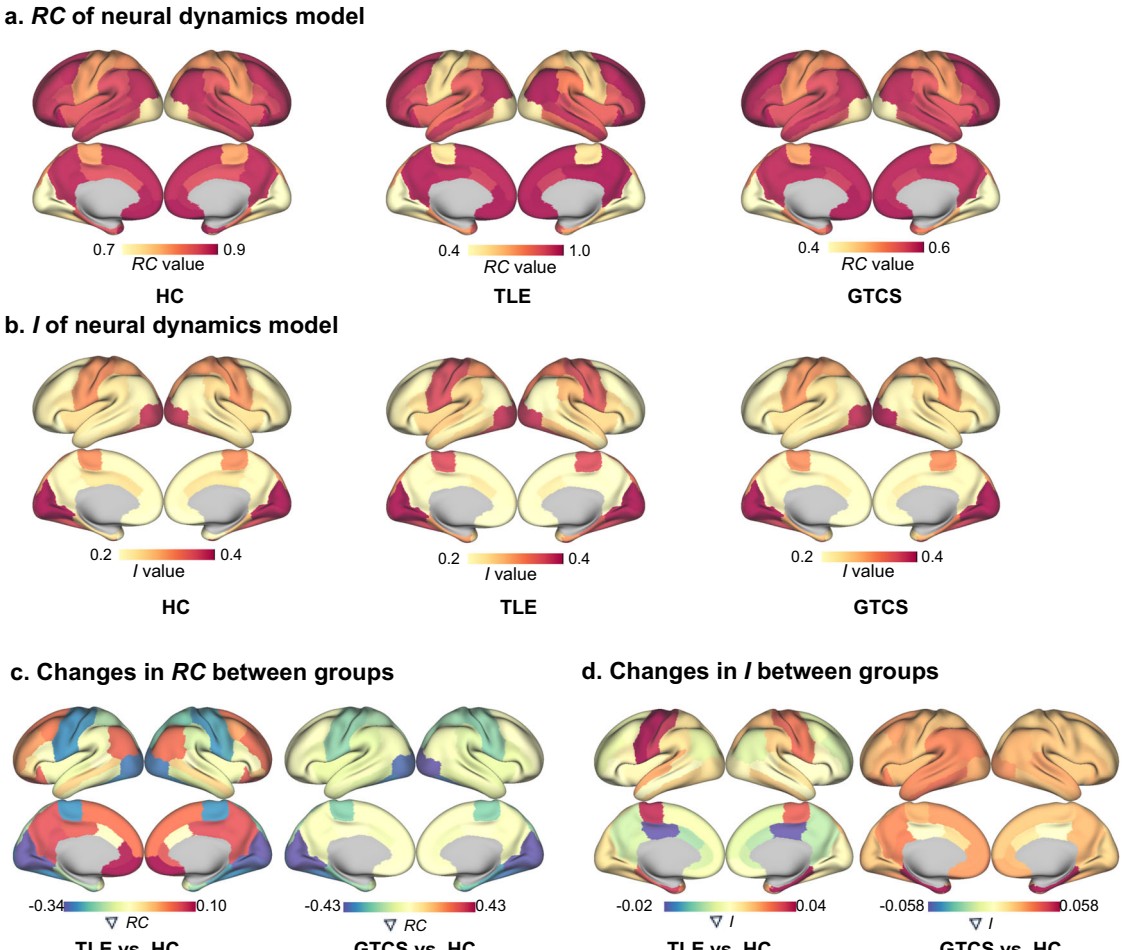

**Fig. 4 Parameters of neural dynamics model in each group.** The spatial pattern of the recurrent connection (*RC*) (**a**) and external input (*I*) (**b**) in HC, TLE and GTCS. Changes in *RC* (**c**) and *I* (**d**) between epilepsies and controls.

correlation of the group differences in both characterizations, rather than the parameters and gradients in each group. We corrected for spatial autocorrelation on both hemispheres using BrainSMASH (Brain Surrogate Maps with Autocorrelated Spatial Heterogeneity)[32]. In BrainSMASH, spatial autocorrelation in brain maps is operationalized through the construction of a variogram. The variogram quantifies, as a function of distance d, the variance between all pairs of points spatially separated by d. Strongly autocorrelated brain maps exhibit less variation in regions with small d than with large d, and are therefore characterized by positive slopes in their variograms. To generate spatial autocorrelation-preserving surrogate brain maps, BrainS-MASH produces random maps whose variograms are approximately matched to a target brain map's variogram. We found significant negative correlation between the differences of PC1 and the changes of RC in TLE ($\rho = -0.72$, $P < 0.001$, Spearman correlation with spatial autocorrelation correction) and GTCS ($\rho = -0.80$, $P < 0.001$, Spearman correlation with spatial autocorrelation correction) (Fig. 5).

In addition to testing the realistic of the model, we examined the association between the gradients of the empirical BOLD signal and gradients of simulated BOLD signal derived from the pMFM. Consistent with the analysis of empirical BOLD signal, we extract massive time-series features of BOLD fMRI simulated from the pMFM by hctsa toolbox, and computed the gradients (PC1 and PC2) of the temporal features using principal component analysis. The spatial patterns of gradients in HC,

TLE and GTCS were shown in Supplementary Fig. 2. We observed similar patterns between the simulated gradients and the empirical gradients in all groups (for PC1, HC: $\rho = 0.319$, $P = 0.008$, TLE: $\rho = 0.526$, $P < 0.001$, GTCS: $\rho = 0.502$, $P < 0.001$; for PC2, HC: $\rho = 0.794$, $P < 0.001$, TLE: $\rho = 0.731$, $P < 0.001$, GTCS: $\rho = 0.726$, $P < 0.001$). Furthermore, the groups comparisons showed spatial similarity between simulated PC1 and empirical PC1 (TLE vs. HC: $\rho = 0.39$, $P = 0.001$; GTCS vs. HC: $\rho = 0.59$, $P < 0.001$). Together, these findings reveal a highly association between the macroscale intrinsic dynamics and microcircuit function in epilepsies.

**Control analyses.** We replicated our results with a higher resolution parcellation with 200 cortical regions[33]. Consistent with our main results, we found that the PC1 shows a ventromedial-lateral gradient and the PC2 captures the sensory-association gradient (Supplementary Fig. 3A), and the RC increased from sensory-motor to association cortices and the I decreased from sensory-motor to association cortices (Supplementary Fig. 3B). For the associations in multiscale brain function, we found negative correlation between the differences of PC1 and the changes of RC in TLE ($\rho = -0.22$, $P = 0.0015$, Spearman correlation with spatial autocorrelation correction) and GTCS ($\rho = -0.27$, $P < 0.001$, Spearman correlation with spatial autocorrelation correction) (Supplementary Fig. 3C).

In addition to the analysis of cortical cortex, and considering that some subcortical regions have been implicated in TLE, like

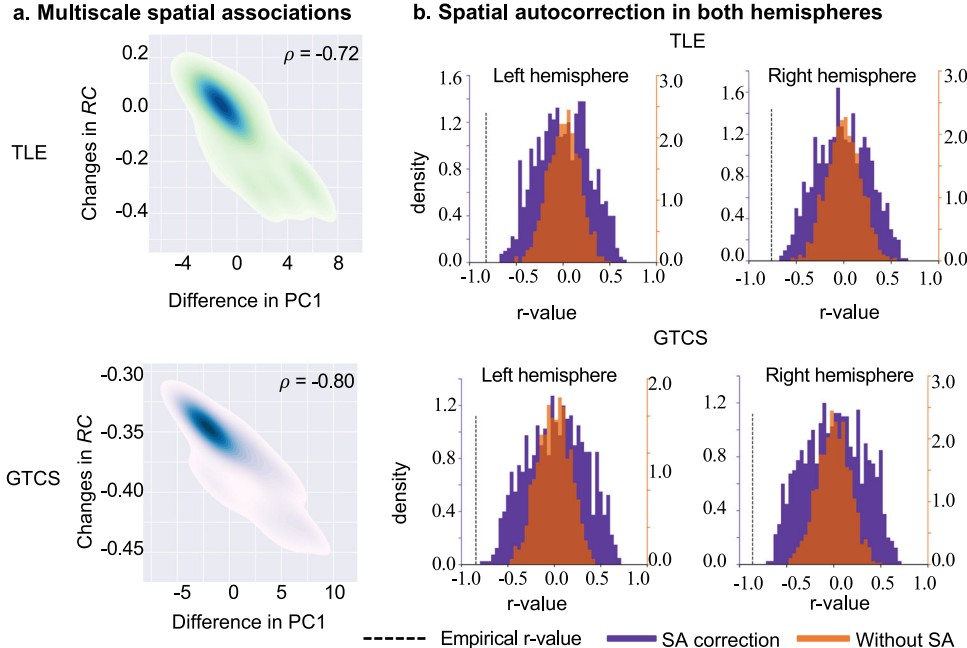

**Fig. 5 Associations in multiscale brain function. a** Negative correlations between the differences in PC1 and changes in RC in TLE group ($\rho = -0.72$, $P < 0.001$; $n = 68$ represented the number of brain parcels) and GTCS ($\rho = -0.80$, $P < 0.001$; $n = 68$ represented the number of brain parcels). For visualizing, the distribution of observations in each brain parcel used a kernel density estimate method. **b** Null distributions of Pearson correlations between changes in RC and 1,000 randomly shuffled (orange) and spatial autocorrection-preserving (purple) surrogate maps, derived from the map of differences in PC1. Dashed black line indicates the empirically observed correlation.

the hippocampus, amygdala, piriform cortex, we performed a supplementary analysis using the masks with subcortices[34]. The PC1 after Procrustes alignment showed an anterior-posterior gradient, and the PC2 changed from the central side to forward and back (Supplementary Fig. 4A). For the comparison between patients and controls, TLE and GTCS showed more shared alterations in PC1, such as the bilateral amygdala and the nucleus accumbens. The two epileptic subtypes showed specific alterations in PC2, such as the bilateral nucleus accumbens in TLE, and the right putamen and the left anterior thalamus and the nucleus accumbens in GTCS (Supplementary Fig. 4B).

## Discussion

In the current study, we explored the associations between macroscale intrinsic dynamics and microcircuit function in focal and generalized epilepsy. Specifically, we found that the differences were observed in the first gradient of time-series features in the primary network and default mode network in TLE and GTCS. Microcircuit recurrent connections were reduced in somatomotor cortices in TLE and GTCS, and were more widely reduced in the latter. The somatomotor cortices are integrated into a multimodal network associated with sensory, motor systems, and cognitive hub regions[35]. By leveraging computational simulations that quantify patterns of information flow across the connectome, TLE showed altered communication dynamics between the somatomotor cortices and subcortical regions, and TLE also showed slower signal spreading time in high-order network with impairment in sensorimotor, executive, and memory functions[36]. More specifically, GTCS has shown abnormal intrinsic brain activity in the primary network, as well as in internetwork communication frameworks. Electroencephalogram–fMRI in patients with idiopathic generalized epilepsy demonstrated that increased connectivity with sensorimotor regions before the onset of discharges can trigger generalized spike wave discharges, involving network changes in the prefrontal cortex and precuneus[37]. Together, the somatomotor

cortices are implicated in two epileptic subtypes. From a multi-scale perspective, our results emphasize the impact of microcircuit abnormal neuronal activity on the primary network and high-order networks, suggesting a systematic abnormality of brain hierarchical organization.

There are differences between intrinsic dynamics gradient derived from the massive features of time-series and the canonical functional connectivity gradient[5]. The functional connectivity gradient focuses on the correlation between regional time-series and then projects it onto a low-dimensional space, describing the hierarchical topography of regional connectivity patterns[38]. The intrinsic dynamics gradient focuses on the massive features of the time-series itself, capturing spatial transitions in regional signal properties. Despite, we further assessed the correlation between intrinsic dynamics gradient and functional connectivity gradient, and found that PC1 did not correlate with the functional connectivity gradient ($\rho = 0.018$, $P = 0.884$), but PC2 was negatively correlated with functional connectivity gradient ($\rho = -0.945$, $P < 0.001$) in health. To compare the intrinsic dynamics gradients with other simple time series properties, we computed the amplitude of low-frequency fluctuation (ALFF) which provided direct characterization of the spontaneous brain activity, and temporal entropy which provide a simple complexity measure of regional time series (Supplementary Fig. 5). After dividing by the global mean ALFF, we found higher mALFF were mainly at the posterior cingulate cortex, precuneus, and the medial prefrontal cortex[39,40]. We then assessed the correlation and found that PC1 ($\rho = 0.469$, $P < 0.001$) was significantly correlated with the mALFF, and PC2 was weakly correlated with mALFF ($\rho = 0.240$, $P = 0.05$). Contrary to the results of mALFF, temporal entropy did not correlate with the PC1 ($\rho = -0.133$, $P = 0.277$), but significantly correlated with the PC2 ($\rho = 0.639$, $P < 0.001$). Together, these time series properties (like the ALFF and entropy) cannot fully capture all aspects of the temporal features of a time series. Rather than selecting conventional or specific time-series properties, the advantage of using hctsa is obtaining massive

temporal features to get a summary characterization. Each feature captures a specific temporal property of regional time-series, not just temporal correlation between paired regions (i.e., functional connectivity). Spontaneous intrinsic dynamics gradient has been shown to be related to other brain organization, such as the first component of gene expression, T1-weighted/T2-weighted, and cortical thickness, suggesting that the macroscale intrinsic dynamics gradient is constrained by the structure of the underlying multiscale organization[41]. Gradient mapping projects the high-dimensional connectome into a manifold space, helping us to understand the pathology of epilepsy from a hierarchical perspective. For example, previous findings demonstrated an extended gradient in patients with GTCS, showing a decrease in the gradient in the somatomotor cortices and an increase in the default mode network[42], which implies the excessive functional segregation between unimodal and transmodal systems. Another study found that dysconnectivity of limbic cortices at the top of the hierarchy was associated with abnormal myelin content and hippocampal atrophy in TLE, dysconnectivity of sensorimotor cortices at the bottom of the hierarchy scale with cortical atrophy[43]. The microstructural gradient reflected asymmetry and atrophy of cortical pattern in TLE, providing complementary insights into the pathology of TLE[44].

Intrinsic dynamics is shaped by microscale attributes and macroscale connectome architecture. The PC1 of time-series features was associated with the autocorrelation properties of BOLD signals, and the PC2 of time-series features was associated with the distribution shape of time-series amplitude[41]. Auto-correlation reflects the periodicity of signals, which may explain the association between the differences in PC1 and the changes in recurrent connection in epilepsies. Similar differences in PC1 in TLE and GTCS, showing increased activity in sensory-motor cortices and decreased in default mode network. The differences in PC1 in TLE were correlated with the stability of time-series, and in GTCS were correlated to the model prediction, suggesting that neuronal activity in the primary network and default mode network was sensitive to the temporal dependencies of time-series in epilepsies. Together, the stability and periodicity of brain activity were abnormal in both epileptic subtypes, but were more prominent in the default mode network in TLE and visual network in GTCS.

The advantage of the pMFM is that it can simulate the neuronal dynamics parameters by combining functional connectivity gradient, which allows it to generate more realistic static and dynamic functional connectivity. The spontaneous activity of the brain at rest, as well as the task-related activity, depends on the underlying structural connections and characteristics of intrinsic dynamics[45]. We noted that the spatial pattern of the changes in recurrent connection followed the hierarchical organization of the brain, from primary network to default mode network in TLE and GTCS. This implies that brain damage was not constrained within the seizure lesions, but rather widespread functional network[15]. We also found that the significant differences in recurrent connection were more prominent in GTCS than in TLE. This is similar to the electroencephalogram pathology of the two epileptic subtypes, suggesting that generalized discharge has a greater impact on the stability of intra-regional connections.

A strong correlation was found between the differences in macroscale intrinsic dynamics gradient and microcircuit changes in recurrent connection in TLE and GTCS. This further indicates that the dysfunctions at different scales interact, as the brain is a multi-scale whole. While intrinsic dynamics gradient and functional connectivity gradient were fundamentally both based on low dimensional representations of the fMRI data. And we inform the biophysical model using the functional connectivity gradient (Fig. 1) to then study how the model parameters of relate

to intrinsic dynamics gradient (Fig. 5). To avoid circular analysis, we focused on the spatial correlation of the group differences in intrinsic dynamics gradient and microcircuit parameters, rather than the parameters and gradient itself in each group. More interesting, the spatial pattern of changes in intrinsic dynamics followed the cortical gradient from sensory-motor cortices to high-order cortices, emphasizing the importance of integration and segregation of brain connectome[46].

There are several limitations in the current study. Second, even though time delays due to axonal propagation can be of crucial important for the study of the whole-brain dynamics, here we assumed instantaneous interactions between the brain regions and assumed that their impact to be encompassed in the neural masses, and we neglected them[9]. Second, the pMFM were derived through a series of major simplifications to arrive at simple, low-dimensional models of neural dynamics, and the results may depend on the exact model form. A data-driven approach that would allow us to avoid choosing a specific neural mass model and instead extract this model directly from the functional data[47]. More important, this data-driven approach can estimate the link between dynamically relevant parameters and the measurable from a preexisting patient cohort, and then applied to a single subject, that might be more helpful for individualized patient estimation. Third, we did not collect the T2 weight data, so we could not calculate T1-weighted/T2-weighted as the structural gradient in the pMFM. Fourth, we used the Desikan–Killiany parcellations to extract the time-series signal and construct the structural connectome. Considering the computing efficiency of the hctsa toolbox and the pMFM, we did not include other atlases with more numbers of cortical regions for validation analyses. Fifth, our findings may have been confounded by the effects of medication on intrinsic dynamics.

In summary, the dynamics changes of brain function at the macroscale and microcircuit in epilepsies highlight the importance of the primary network and default mode network in the cortical hierarchy, and expand our understanding of the neural mechanisms of disease from the perspective of System Epilepsies hypothesis.

## Methods
**Participants**. The study included 75 patients with temporal lobe epilepsy (TLE), 79 patients with genetic generalized epilepsy with generalized tonic-clonic seizures (GTCS), and 108 healthy controls (HC) (Table 1). All study protocols were performed according to the Helsinki Declaration of 1975 and approved by the medical ethics committee of Jinling Hospital, School of Medicine, Nanjing University. Written informed consent was obtained from all participants.

Two types of patients had negative presentation on diagnostic MRI (i.e., no lesion such as cortical dysplasia, benign brain tumor or hippocampal sclerosis was detected), and none had received surgical treatment up to time of data preparation for this work. According to International League against Epilepsy (ILAE) classifications[48], epilepsies were diagnosed by two experienced neurologists. For patients with TLE, the patients present with (a) typical symptoms of TLE, such as automatism (hand, oral), autonomic symptoms, olfactory hallucination, and complex partial seizures with aura; and (b) specific patterns of electrophysiological activity recorded by scalp electroencephalogram, such as epileptic spikes in the bilateral frontotemporal or temporal lobes. For genetic generalized epilepsy with GTCS, the patients presented with a) typical seizure semiology of GTCS, including loss of consciousness during seizures without precursory symptoms of partial epilepsy and aura, and tonic extension of the limbs followed by a clonic phase of rhythmic

**Table 1 Demographic and clinical characteristics of participants.**

| Demographics | HC | TLE | GTCS | Comparisons | |
|---|---|---|---|---|---|
| | (n = 108) | (n = 75) | (n = 79) | Statistical value | P value |
| Sex (males/females) | 59/49 | 43/34 | 54/25 | $\chi^2 = 0.03$ (P = 0.87) | $\chi^2 = 3.59$ (P = 0.06) |
| handedness (left/right) | 0/108 | 0/75 | 0/79 | $\chi^2 = 0$ (P = 1) | $\chi^2 = 0$ (P = 1) |
| Age (years) | 23.72 ± 4.15 | 25.72 ± 8.64 | 24.78 ± 7.84 | U = 3889 (P = 0.64) | U = 3769 (P = 0.17) |
| Duration of illness (months) | — | 111.90 ± 89.25 | 87.27 ± 93.71 | — | — |
| Seizure onset age (years) | — | 16.51 ± 8.71 | 17.88 ± 7.65 | — | — |
| Treatment (yes/no) | — | 55/9[a] | 51/19[a] | — | — |
| Seizure frequency (month) | — | 6.06 ± 19.50 | 17.46 ± 80.88 | — | — |

*HC healthy controls, TLE temporal lobe epilepsy, GTCS genetic generalized epilepsy with generalized tonic-clonic seizures.*
Values are mean ± standard deviation (SD).
The $\chi^2$ value for gender distribution was obtained by chi-square test.
The U values were obtained by Mann-Whitney tests.
[a]Missed clinical information

jerking, b) generalized spike-and-wave discharges on electroencephalogram, and c) no other epilepsy associated etiology such as trauma, tumor, or intracranial infection. All patients had negative presentation on conventional (structural) magnetic resonance images (MRI), and a portion of patients had concordant positron emission tomography or magnetoencephalogram examination. Moreover, patients were excluded for i) progressive diseases, malformations of cortical development, tumors, or previous neurosurgery, iii) incomplete MR scanning, or iv) excessive head motion during scanning.

**Imaging protocol**. Functional and structural images were acquired on a Siemens Trio 3.0-tesla MRI scanner at Jinling Hospital. We used foam padding to minimize head motion. All participants were required to keep their eyes closed and not to fall asleep. Functional MR images (fMRI) were acquired using an echo-planar sequence (repetition time = 2000 ms, echo time = 30 ms, and flip angle = 90°). Thirty-three transverse slices (field of view = 240 × 240 mm$^2$, matrix = 64 × 64, slice thickness = 4 mm, and interslice gap = 0.4 mm) aligned along the anterior commissure–posterior commissure line was acquired with a total of 250 volumes. Total scan time was 500 s. T1-weighted MRI were acquired in a sagittal orientation using a magnetization-prepared rapid gradient-echo sequence (repetition time/echo time = 2,300/2.98 ms, flip angle = 9°, field of view = 256 × 256 mm$^2$, matrix size = 256 × 256, slice thickness = 1 mm, no interslice gap, and 176 slices). Diffusion weighted images were obtained using spin an echo-based echo planar imaging sequence, including 30 volumes with diffusion gradients applied along 30 non-collinear directions (b = 1,000 s/mm$^2$) and one volume without diffusion weighting (b = 0 s/mm$^2$). Each volume consisted of 45 contiguous axial slices (repetition time/echo time = 6,100 ms/93 ms, flip angle = 90, field of view = 240 × 240 mm$^2$, matrix size = 256 × 256). All participants were asked if they had fallen asleep during the scanning.

**Data preprocessing**. Functional images were preprocessed using the DPABI toolbox[49] (http://rfmri.org/dpabi) which synthesizes procedures in SPM12 software[50] (http://www.fil.ion.ucl.ac.uk/spm). The first ten images were discarded, and the remaining images were corrected for temporal differences and head motion, spatially normalized to Montreal Neurologic Institute space, and re-sampled to 3 × 3 × 3 mm$^3$ voxels. The nuisance variables including the Friston 24-parameter model, ventricular signal, and white matter signal were regressed out. Frame-wise displacement was calculated for each time point. Participants were excluded if any of the following three criteria were satisfied, (i) mean frame-wise displacement exceeded 0.5 mm, (ii) head motion exceeded

3 mm or 3°, or (iii) more than 20% of all time points had frame-wise displacement values exceeding 0.5 mm. Functional images were then spatially smoothed with an 8-mm full-width at half-maximum isotropic Gaussian kernel. Finally, linear trends were removed from the signal, and temporal band-pass filtering (0.01–0.08 Hz) was performed.

Diffusion weighted images were corrected using a non-diffusion-weighted B$_0$ image and a filed map for accounting for the eddy-current-induced distortions and reduced head movements through FSL. Then structural connectomes were generated from preprocessed diffusion weighted images using MRtrix3[51]. The T1-weighted images were segmented into different tissue types, including cortex and subcortical gray matter, white matter, and cerebrospinal fluid. The T1-weighted was registered to the diffusion weighted images using the FLIRT, and the transformation was applied to different tissue types to register them to the native diffusion weighted images space. We estimated the multi-organization response function and perform constrained spherical deconvolution and intensity normalization. All white matter voxels were taken as seed points, and iFOD2, the default algorithm of toolkit, was used for streamline tracking. Subsequently, spherical deconvolution informed filtering of tractograms (SIFT2) algorithm was used to optimize the cross-sectional magnification of each streamline to match the whole brain fiber bundle map. Finally, streamlines were mapped to the Desikan–Killiany atlas with 68 cortical regions[29], and then log transformation[52] was used to construct the structural connections used as model inputs.

**Computation of macroscale intrinsic dynamics**. To describe the macroscale intrinsic dynamics, we comprehensively extracted massive temporal features of time-series derived from BOLD fMRI. We first extracted regional time-series across cerebral cortex for each participant using Desikan–Killiany anatomical parcellation. We additionally replicated the results with a higher resolution parcellation with Schaefer 200 cortical regions in controls analysis (Supplementary Fig. 3). We then used highly comparative time-series analysis (hctsa) toolbox, a computational framework for automated time series using massive feature extraction[30], upon which allows to transform each time-series to a set of over 7,000 features that each encode a different scientific analysis method. These time-series features described temporal properties including distributional properties, entropy and variability, autocorrelation, and nonlinear properties of a given time-series[53]. Following the feature extraction procedure, we removed the errored outputs of the operations, and the remaining shared features (5,360) among three groups were normalized across regions using an outlier-robust sigmoidal transform. As a

result, a feature matrix (5360 × 68) was constructed for each participant, representing the temporal fingerprints of cortical regions. To investigate the topographic organization of macro-scale intrinsic dynamics, we identified the spatial gradient of temporal features. We therefore performed the principal component analysis to capture spatial gradients that explained maximal variance. We mainly focused on the first component (PC1) and the second component (PC2) (Fig. 1).

For the analysis of subcortical regions, using the same steps as in the cortical analysis, we analyzed the macroscopic intrinsic dynamics in the subcortical regions. To extract the time-series, we used the Tian subcortical masks[34] which contain 16 subcortical structures. We computed massive temporal features using the hctsa toolbox. Following the feature extraction procedure, a shared feature matrix was constructed for each participant, representing the temporal fingerprinting of subcortical regions. We then performed the principal component analysis to identify the spatial gradient, and compared the PC1 and PC2 in patients with epilepsy to the healthy controls.

**Microcircuit simulation by pMFM.** To describe the microcircuit function of neural population, we used a large-scale biophysically plausible model, the parametric mean field model (pMFM)[28], to simulate local neural dynamics, including the recurrent connection (RC) and external input (I). In the pMFM, the neural dynamics of each cortical region are driven by four components, (1) intra-regional input, (2) external input, (3) inter-regional input, and (4) neuronal noise. Specifically, intra-regional input and external input correspond to the free parameters, the RC and I, respectively. Inter-regional inputs are parameterized by the structural connectivity and scaled by a global scaling constant G. Neuronal noise is assumed to be Gaussian with a standard deviation σ. The nonlinear stochastic differential equations of neural activity in each cortical region followed the same parameter settings as in the previous study[28].

The simulated neural activities were fed to the Balloon-Windkessel hemodynamics model[54,55] to simulate the BOLD fMRI. To reduce the calculation time, Euler's integration with a time step of 50 ms was used to simulate the pMFM and the hemodynamic model in the training process, but a time step of 10 ms was used in the validation and test process to generate realistic signals. Since the duration of empirical fMRI data collected was 8 minutes, the duration of simulated fMRI data was set to be 10 minutes. Then the first 2 minutes was discarded, and the time-series is downsampled to 2 seconds to maintain the same time-interval resolution as our empirical data.

In the pMFM, the parameters RC, I, and σ were set as a linear combination of the FC gradient, resulting 7 unknown coefficients: G and $a_w$, $b_w$, $a_I$, $b_I$, $a_\sigma$, $b_\sigma$:

$$RC_i = a_w FCG_i + b_w \quad (1)$$

$$I_i = a_I FCG_i + b_I \quad (2)$$

$$\sigma_i = a_\sigma FCG_i + b_\sigma \quad (3)$$

Here, $RC_i$, $I_i$, $\sigma_i$ represented the recurrent connection, external input and neuronal noise of the i cortical region, and the FCG represented the principal gradient of functional connectivity. We did not consider the T1-weighted /T2-weighted myelin property because we did not collect the T2w data.

Our goal was to estimate the microcircuit function, the recurrent connection and external input. To optimize the pMFM, we estimated the 7 unknown coefficients by minimizing the disagreement between the empirical and simulated FC and functional connectivity dynamics (FCD) matrices. The agreement between empirical and simulated 68 × 68 FC was estimated by

maximizing the Pearson correlation between the two matrices. The FCD was generated by computing the correlation between a sliding window, with 1TR as the stepwise, yielding a 211 × 211 FCD matrix for each participant. The agreement between empirical and simulated FCD was estimated by minimizing the Kolmogorov-Smirnov distance between the probability distribution functions (pdfs) constructed from the upper triangular entries of the two FCD matrices. The pdf of an FCD matrix was constructed by collapsing the upper triangular entries of the matrix into a histogram and normalizing it to have an area of one. The holistic optimization minimized the cost function, implying a better fit between empirical and simulated FC and FCD. According to a previous study[28], we applied covariance matrix adaption evolution strategy algorithm to estimate cost function (Fig. 2).

The HC, TLE and GTCS groups were divided into three subsets: training set, validation set and test set. Participants were demographic and clinical matched in each subset (Supplementary Table 1). Given a particular random initialization of the 7 unknown coefficients, the covariance matrix adaption evolution strategy algorithm was applied to the training set. Each algorithm was iterated 500 times, and this procedure was repeated 10 times, then generated 5,000 coefficients sets. The 5,000 coefficients sets were evaluated in the validation dataset, and the top 10 coefficients sets with the lowest cost in validation procedure were selected to apply to the test dataset. We sought to determine the top 1 coefficient set that show the lowest cost in test procedure, and finally confirm the best coefficient set for further analyses. According to the best coefficient set, we estimated the neural dynamics (RC and I) in HC, TLE and GTCS groups.

**Association analysis in multi-scale dynamics.** We aimed to explore the associations between the macroscale intrinsic dynamics and microcircuit function. First, we directly assessed the spatial correlation between the differences of gradients (PC1 and PC2) of time-series features and changes of neural dynamics (RC and I) in epilepsies. To this end, we computed the Spearman correlation between these maps, and further corrected the spatial autocorrelation[32] on both hemispheres. We then tested the similarity of gradients between empirical data and simulated data generated by the pMFM. Consistent with the analysis of empirical BOLD signal, we used the hctsa toolbox to extract time-series features of simulated BOLD fMRI generated by the pMFM, and computed the gradients (PC1 and PC2) of time-series features. We also computed the Spearman correlation between the empirical gradients and simulated gradients in three groups.

**Statistics and reproducibility.** To determine the differences of macroscale intrinsic dynamics in epilepsies, we compared the gradients (PC1 and PC2) of time-series features among individuals in three groups. First, to align the components among individuals, we constructed group-level intrinsic dynamics gradients using the data from healthy controls, and then used the Procrustes method[56] to align the individual gradients to the group-level gradients. To compare the group difference in gradients, we then performed 5,000 permutation tests ($P_{FWE} < 0.05$) on the left and right hemispheres respectively through the FSL toolkit PALM. Age and sex were regressed out.

To examine the microcircuit dysfunction in epilepsies, we analyzed changes defined by subtracting between patients and controls, since the RC and I were representative of the group-average level. Then we performed a total of 1,000 permutation tests ($P_{FDR} < 0.05$) to determine the regions with significantly statistical changes, by randomly shuffling the epilepsies and

controls within training, validation and test sets, and then repeated the same analysis in the pMFM procedure.

**Reporting summary**. Further information on research design is available in the Nature Portfolio Reporting Summary linked to this article.

## Data availability

Processed neuroimaging data from Jinling Hospital have been deposited at Figshare, https://doi.org/10.6084/m9.figshare.24966507.v1. The raw data that support the findings of this study are available on request from the corresponding author. The source data used to plot Figures and Supplementary Figures can be found in the Supplementary Data 1 and 2 respectively.

## Code availability

All the code is openly available at https://github.com/SiqiYang47/Multi-scale-brain-function. The code of pMFM is avialable at https://github.com/ThomasYeoLab/CBIG/tree/master/stable_projects/fMRI_dynamics/Kong2021_pMFM.

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

## Acknowledgements

We are grateful to all the participants in this study. This study was funded by the National Natural Science Foundation of China (82302293 and 61871077), the Science and Technology foundation of Sichuan Province (24NSFSC5342), and the Talent Introduction Research Projects of Chengdu University of Information Technology (KYTZ2023036).

## Author contributions

Conceptualization, Supervision, Project Administration and Funding Acquisition, W.L. and H.C.; Formal Analysis, Y.M., and C.P.; Methodology, S.Y., X.K., and S.Z.; Resources, Z.Z.; Software: R.K., and S.Y.; Visualization, S.Y., and T.Y.; Validation, S.Y., and Y.Z.; Writing – Original Draft, S.Y., and T.Y.; Writing – Review & Editing, S.Y., and T.Y.

## Competing interests

The authors declare no competing interests.
