## [Peer Review File · Communications Biology]

Reviewers' comments:

Reviewer #1 (Remarks to the Author):

In the manuscript entitled "Associations between Macro- and Microscale Functional dynamics in Focal and Generalized Epilepsies" by Yang et. al, the authors present a manuscript analyzing fMRI data and macroscale neural mass dynamics in two subtypes of patients with epilepsy. Specifically, the authors use a toolbox called highly-comparative timeseries analysis (hctsa) to extract ~5,000 timeseries features from fMRI data, and collapsed those features using PCA. The authors also used a mean field model (MFM) to simulate neuronal dynamics. It appears that the fundamental assertion of this paper is that the authors have conducted a comparison between macroscale dynamics (via fMRI) and microscale dynamics (via the MFM) in epilepsy. However, while the MFM does include simulation of excitatory and inhibitory neural populations, it is still fundamentally a model of macroscale dynamics, not microscale dynamics. Thus, the authors' analysis amounts to an analysis of macroscale (observed and simulated) dynamics only. Additionally, it's not entirely clear to me that the authors use of hctsa is warranted. The purpose of hctsa is to extract high-dimensional time series features and examine their contribution to a research question. Here however, the authors collapsed their ~5,000 hctsa features down to 2 principal components (PCs), eschewing the advantage of using hctsa. It seems to this reviewer that the authors could have easily extracted their 2 PCs from simple first-order connectivity estimates (cf. Margulies). Unfortunately, it's not clear what hctsa adds here. Lastly, the authors use their PCs to inform the parameter estimates of the MFM and then find that the PCs and MFM parameters correlate strongly. The authors take this as evidence of their goal to link microscale and macroscale dynamics in epilepsy. However, these effects appear trivial given the circularity embedded in their approach (i.e., using PC to inform MFM estimation followed by comparison of PC to MFM params). I think more detail is needed in these analyses (including how the spatial autocorrelation nulls were implemented to avoid this circularity issue) since these correlations appear to be central to the authors' goals (i.e., linking micro to macroscale dynamics).

Overall, unfortunately I did not find the authors' analyses to be a compelling test of their research aims.

Minor comments

- Numerous spelling and grammar issues throughout the manuscript.
- Figure 5 has boxes instead of r/ρ values in sub panels

Reviewer #2 (Remarks to the Author):

This is an interesting manuscript with exciting methodological techniques to explore network changes in epilepsy.

My comments are outlined below – I hope the authors find these useful.

- I was surprised to see a cortical parcellation mask in a study where TLE is a primary focus, as the main regions implicated in TLE are the hippocampus, amygdala, piriform cortex, etc. It would be worth re-analyzing the data with sub-cortical masks (e.g., <https://github.com/yetianmed/subcortex>). The results may be more neurologically valid if these regions are included in the multimodal network analysis.

- Also, most often, TLE and GTCS have a laterality effect (seizures stats with one hemisphere). It is

worth considering analysing ipsilateral vs. contralateral hemispheres (by flipping hemispheres) rather than left vs. right.

- Table 1 needs to include a lot of clinical information. For example.

- o Seizure onset age
- o Laterality of seizure
- o Seizure burden
- o Medication history
- o Specific EEG findings

- Was head motion different between any of the sub-groups?

- What subtypes of Genetic Generalised Epilepsy did subjects with GTCS have?

- The authors state that: "All patients had negative presentation on conventional (structural) magnetic 362 resonance images (MRI), and a portion of patients had concordant positron emission 363 tomography or magnetoencephalogram examination". Does this mean that GTCS and TLE were lesion-negative? This should be highlighted further in the manuscript because this isn't a more conventional hippocampal sclerosis group.

Reviewer #3 (Remarks to the Author):

Authors study changes in macroscopic functional connectivity, as well as in parameters of the mesoscale mean field models in two types of epilepsy, as compared with a control group. The methodology seems solid and follows the state of the art, the size of the groups is also decent, but the results are less clear and better contextualizing them would improve the work. For example, is there any reason why the somatomotor regions would be mostly impacted? Similarly, from the beginning is not clear why the emphasize is on the gradient, instead of for example RSNs, or simply staying at nodes' level.

Another weak point is the quite coarse parcelation that is used. At least for TLE, Virtual Epileptic Brain (VEP) atlas seems to offer better distinction between the regions of interest. The results would have been much strengthened if the analysis is confirmed in this, or in some other finer parcelation.

Other comments:

- In the introduction, the authors should mention the work by Courtiol et al. J neurosci 2020, and possibly other works that analyzed the rs fMRI in epileptic patients, such as Wirsich et al. Neuroimage 2016.

- instantaneous interactions are assumed between the brain regions, even though time-delays due to axonal propagation can be very important for the whole-brain dynamics, e.g. for synchronization. This needs to be justified and mentioned in the limitations.

- In the Limitations it is worth mentioning about the possible limitations of the neuronal masses in general, which can be overcome by data-driven brain network models, such as one by Sip et al. Sci Adv 2023.

- figures are not very sharp (especially bad are 2 and 3), vector graphics should be used if possible.

- l450 there seems to be a missing word in the sentence

Response to Reviewers (COMMSBIO-23-2881A)

=====

Reviewer #1:

In the manuscript entitled “Associations between Macro- and Microscale Functional dynamics in Focal and Generalized Epilepsies” by Yang et. al, the authors present a manuscript analyzing fMRI data and macroscale neural mass dynamics in two subtypes of patients with epilepsy. Specifically, the authors use a toolbox called highly-comparative timeseries analysis (hctsa) to extract ~5,000 timeseries features from fMRI data, and collapsed those features using PCA. The authors also used a mean field model (MFM) to simulate neuronal dynamics. It appears that the fundamental assertion of this paper is that the authors have conducted a comparison between macroscale dynamics (via fMRI) and microscale dynamics (via the MFM) in epilepsy.

Response:

We greatly appreciate the insightful comments on our manuscript. We made modifications according to the suggestions and included point-by-point responses to each issue.

Q1: However, while the MFM does include simulation of excitatory and inhibitory neural populations, it is still fundamentally a model of macroscale dynamics, not microscale dynamics. Thus, the authors’ analysis amounts to an analysis of macroscale (observed and simulated) dynamics only.

Response:

We agree with the reviewer’s viewpoint. In the old version of the manuscript, the reason why we used microscale dynamics to describe neural dynamics (recurrent connection (*RC*) and external input (*I*) simulated by the pMFM) is based the following main considerations. The pMFM can potentially provide insights into the microscale organization of the dynamic resting brain by simulating large-scale biophysical models. The dynamical properties of these large-scale circuit models are governed by parameters with physical interpretation. Some of these parameters can be empirically measured using cellular neurophysiology or histology. Thus, compared to the gradient of time-series features, the simulated neural dynamics (*RC* and *I*) were considered to be brain microscale functional properties.

We recognized that there were some drawbacks to using the term "microscale dynamics". Based on a related study (Park et al., 2021), we used microcircuit function to describe the parameters modelled from pMFM, and modified relevant statements in the revision as follows. If there is still something inappropriate, we are happy to make further changes.

(Page 2) “However, it is not clear to what extent the epilepsy-related perturbations of spontaneous brain activity affect macroscale intrinsic dynamics and microcircuit organizations, that supports their pathological relevance. We collected a sample of patients with temporal lobe epilepsy (TLE) and genetic generalized epilepsy with tonic-clonic seizure (GTCS), as well as healthy controls. We extracted massive temporal

features of fMRI BOLD time-series to characterize macroscale intrinsic dynamics, and simulate microcircuit neuronal dynamics used a large-scale biological model. We investigate whether macroscale intrinsic dynamics and microcircuit dysfunction differed in epilepsies, and how these changes are linked.

(Page 2) “Biophysical simulations indicated reduced recurrent connection within somatomotor microcircuits in both subtypes, and even more reduced in GTCS. We further demonstrated strong spatial correlations between differences in the gradient of macroscale intrinsic dynamics and microcircuit dysfunction in epilepsies.”

(Page 3) “Nevertheless, the impact of brain dysfunction caused by abnormal neuronal activity is not fully understood. How this dysfunction affects the spatial arrangement of cortical gradient and macroscale spontaneous brain activity remains unknown.

(Page 5) “In addition, we used a parametric mean field model to simulate microcircuit function, including recurrent connections and external input. We then analyzed the association between these multiscale brain function in epilepsies.”

(Page 5) “Macroscale intrinsic dynamics and microcircuit function”

(Page 12) “Microcircuit simulations by the pMFM”

(Page 13) “To further examine the associations between the macroscale intrinsic dynamics and microcircuit function in epilepsies, we assessed the spatial correlation between the altered gradients of time-series features and changes of the recurrent connections and external input yielded from a neural dynamics model.”

(Page 15) “Together, these findings reveal a highly association between the macroscale intrinsic dynamics and microcircuit function in epilepsies.”

(Page 17) “From a multi-scale perspective, our results emphasize the impact of microcircuit abnormal neuronal activity on the primary network and high-order networks, suggesting a systematic abnormality of brain hierarchical organization.”

(Page 28) “Our goal was to estimate the microcircuit function, the recurrent connection and external input.”

Bibliography in the Response to Reviewers:

Park, B. Y. et al. Differences in subcortico-cortical interactions identified from connectome and microcircuit models in autism. Nat Commun 12, 2225, doi:10.1038/s41467-021-21732-0 (2021)

Q2: Additionally, it's not entirely clear to me that the authors use of hctsa is warranted. The purpose of hctsa is to extract high-dimensional time series features and examine their

contribution to a research question. Here however, the authors collapsed their ~5,000 hctsa features down to 2 principal components (PCs), eschewing the advantage of using hctsa. It seems to this reviewer that the authors could have easily extracted their 2 PCs from simple first-order connectivity estimates (cf. Margulies). Unfortunately, it's not clear what hctsa adds here.

Response:

Thank the reviewer for this valuable suggestion. First, we used the *hctsa* to capture rich temporal signatures because they represent multiple aspects of intrinsic dynamics, not just selected specific time-series features. Second, for each participant, a feature matrix (5,360 × 68) represented the temporal fingerprints of intrinsic dynamics in cortical regions. The PCs obtained by identified the main axes of variance in the matrix represented gradients, along which cortical locations are ordered according to their similarity in temporal features to the rest of the cortex. Thus, we analyzed topographic gradient of intrinsic dynamics in our study.

The intrinsic dynamics gradient and the canonical functional connectivity gradient differ in their approaches to characterizing brain organization. The functional connectivity gradient focuses on the correlation between regional time series, and then projects it onto a low-dimensional space, describing the hierarchical topography of regional connectivity patterns. The intrinsic dynamics gradient focuses on the features of the time series itself, capturing the spatial transitions of regional signal properties. Each feature reflects a specific temporal property of the regional time series, providing a more multi-dimensional representation of brain activity compared to functional connectivity analysis.

Based on the suggestion, we modified statements in the revision as follows (Pages 6, 17, 18):

(Page 6) "From the perspective of dimensionality reduction method, gradient more focused on the relationship between brain regions regard to their similarity of 5,000 shared temporal features, which representing the fingerprints of regional intrinsic dynamics. We therefore emphasized the gradient of temporal features rather than temporal correlation of time-series."

(Pages 17-18) "There are differences between intrinsic dynamics gradient derived from the massive features of time-series and the canonical functional connectivity gradient⁵. The functional connectivity gradient focuses on the correlation between regional time-series and then projects it onto a low-dimensional space, describing the hierarchical topography of regional connectivity patterns³⁸. The intrinsic dynamics gradient focuses on the massive features of the time-series itself, capturing spatial transitions in regional signal properties. The advantage of using hctsa is obtaining massive temporal features. Each feature captures a specific temporal property of regional time-series, not just temporal correlation between paired regions (i.e., functional connectivity).

Q3: Lastly, the authors use their PCs to inform the parameter estimates of the MFM and

then find that the PCs and MFM parameters correlate strongly. The authors take this as evidence of their goal to link microscale and macroscale dynamics in epilepsy. However, these effects appear trivial given the circularity embedded in their approach (i.e., using PC to inform MFM estimation followed by comparison of PC to MFM params). I think more detail is needed in these analyses (including how the spatial autocorrelation nulls were implemented to avoid this circularity issue) since these correlations appear to be central to the authors' goals (i.e., linking micro to macroscale dynamics).

Response:

We apologized for the misleading statements, and we did not use the principal components (PCs) of time series features to inform pMFM parameters estimation. Consistent to a previous study (Kong et al., 2021), the parameters RC , l , and σ were set as a linear combination of the functional connectivity gradient (Margulies) in the pMFM. The time series features gradient is different from the Margulies gradient. The former is based on the regional time-series features matrix, which describes the local aspect of temporal properties, and the latter is based on the temporal correlation between regional time-series.

For the analyses of linking the microcircuit simulations to macroscale dynamics, we corrected the spatial autocorrelation (SA) on both hemispheres using BrainSMASH (Brain Surrogate Maps with Autocorrelated Spatial Heterogeneity, <https://github.com/murraylab/brainsmash/tree/master>) (Fulcher et al., 2021). In BrainSMASH, SA in brain maps is operationalized through the construction of a variogram. The variogram quantifies, as a function of distance d , the variance between all pairs of points spatially separated by d . Brain maps with very little SA will therefore have a variogram which is nearly flat. Strongly autocorrelated brain maps exhibit less variation in regions with small d than with large d , and are therefore characterized by positive slopes in their variograms. To generate SA-preserving surrogate brain maps, BrainSMASH produces random maps whose variograms are approximately matched to a target brain map's variogram. Like the upper left part of the Fig.1 below, dashed black line indicates the empirically observed correlation ($\rho = -0.72$) between the difference of PC1 and the changes of RC in left hemisphere in TLE group. Null distributions of Pearson correlations between changes in RC and 1,000 randomly shuffled (orange) and spatial autocorrection (SA)-preserving (purple) surrogate maps, derived from the map of differences in PC1.

For the spatial autocorrelation, we clarified the relevant statements in the revision as follows (Page 13):

(Page 13) "We corrected for spatial autocorrelation (SA) on both hemispheres using BrainSMASH (Brain Surrogate Maps with Autocorrelated Spatial Heterogeneity)³². In BrainSMASH, SA in brain maps is operationalized through the construction of a variogram. The variogram quantifies, as a function of distance d , the variance between all pairs of points spatially separated by d . Strongly autocorrelated brain maps exhibit less variation in regions with small d than with large d , and are therefore characterized by positive slopes in their variograms. To generate SA-preserving surrogate brain maps, BrainSMASH produces random maps whose variograms are approximately matched to

a target brain map's variogram."

Fig.1 Correction for spatial autocorrelation in both hemispheres. Dashed black line indicates the empirically observed correlation. Null distributions of Pearson correlations between changes in RC and 1,000 randomly shuffled (orange) and spatial autocorrection (SA)-preserving (purple) surrogate maps, derived from the map of differences in PC1.

Bibliography in the Response to Reviewers:

Fulcher, B. D., Arnatkeviciute, A. & Fornito, A. Overcoming false-positive gene- category enrichment in the analysis of spatially resolved transcriptomic brain atlas data. *Nat Commun* 12, 2669, doi:10.1038/s41467-021-22862-1 (2021).

Q4: Numerous spelling and grammar issues throughout the manuscript.

Response:

We tried our best to modify the spelling and grammar throughout the manuscript. The modified text in the manuscript is highlighted in yellow.

Q5: Figure 5 has boxes instead of r/rho values in sub panels.

Response:

We apologized for the mistake. We modified this figure in the revision as follows:

=====

Reviewer #2:

This is an interesting manuscript with exciting methodological techniques to explore network changes in epilepsy. My comments are outlined below – I hope the authors find these useful.

Response:

We greatly appreciate the insightful comments on our manuscript and the opportunity to submit a revision. We made modifications according to the suggestions and included point-by-point responses to each issue.

Q1: I was surprised to see a cortical parcellation mask in a study where TLE is a primary focus, as the main regions implicated in TLE are the hippocampus, amygdala, piriform cortex, etc. It would be worth re-analyzing the data with sub-cortical masks (e.g., <https://github.com/yetianmed/subcortex>). The results may be more neurologically valid if these regions are included in the multimodal network analysis.

Response:

Thank the reviewer for this valuable suggestion. We agree that the regions implicated in TLE are the hippocampus, amygdala, piriform cortex, we performed supplementary analyses using the subcortical masks (<https://github.com/yetianmed/subcortex>).

First, we analyzed the macroscale intrinsic dynamics in the subcortical regions, using the same steps as in the cortical analysis. We used the subcortical masks to extract the time series and computed massive temporal features by the highly comparative time series analysis (*hctsa*) toolbox. We then performed principal component analysis to identify the spatial gradient, which reflected the topographic organization of macroscale intrinsic dynamics. However, for the analysis of microcircuit functional dynamics, the parameter mean field model (pMFM) cannot support simulation of the subcortical neuronal dynamics. The reason might be that in the pMFM, the neural dynamics of each cortical region is driven by four components: (1) recurrent (intra-regional) input, (2) inter-regional inputs, (3) external input (potentially from subcortical relays), and (4) neuronal noise (Wang et al., 2019; Kong et al., 2021). The subcortical regions have been considered for external energy inputs. If the brain parcellation with subcortical masks is reanalyzed, the original assumption of this model will not be satisfied.

The analysis of subcortical masks has been added as follows (Pages 16 and 25):

(Page 16) "In addition to the analysis of cortical cortex, and considering that some subcortical regions have been implicated in TLE, like the hippocampus, amygdala, piriform cortex, we performed a supplementary analysis using the masks with subcortices³⁴. The PC1 after Procrustes alignment showed an anterior-posterior gradient, and the PC2 changed from the central side to forward and back (Fig.S4A). For the comparison between patients and controls, TLE and GTCS showed more shared alterations in PC1, such as the bilateral amygdala and the nucleus accumbens. The two epileptic subtypes showed specific alterations in PC2, such as the bilateral nucleus accumbens in TLE, and the right putamen and the left anterior thalamus and the

nucleus accumbens in GTCS (Fig.S4B)."

(Page 26) *"For the analysis of subcortical regions, using the same steps as in the cortical analysis, we analyzed the macroscopic intrinsic dynamics in the subcortical regions. To extract the time-series, we used the Tian subcortical masks³⁴ which contain 16 subcortical structures. We computed massive temporal features using the htcsa toolbox. Following the feature extraction procedure, a shared feature matrix was constructed for each participant, representing the temporal fingerprinting of subcortical regions. We then performed the PCA to identify the spatial gradient, and compared the PC1 and PC2 in patients with epilepsy to the healthy controls."*

Fig.S1 Spatial pattern of the PC1 and PC2 in each group after Procrustes alignment (A). Network-wise differences in PC1 and PC2 were compared (B). A radar plot shows the difference between the epilepsies and HC. The star represents significant difference between groups, $P < 0.05/16$ with Bonferroni correction.

Bibliography in the Response to Reviewers:

Wang, P. et al. Inversion of a large-scale circuit model reveals a cortical hierarchy in the dynamic resting human brain. *Sci Adv* 5, eaat7854, doi:10.1126/sciadv.aat7854 (2019)

Kong, X. et al. Sensory-motor cortices shape functional connectivity dynamics in the human brain. *Nat Commun* 12, 6373, doi:10.1038/s41467-021-26704-y (2021).

Tian, Y., Margulies, D. S., Breakspear, M. & Zalesky, A. Topographic organization of the human subcortex unveiled with functional connectivity gradients. *Nat Neurosci* 23, 1421-1432, doi:10.1038/s41593-020-00711-6 (2020).

Q2: Also, most often, TLE and GTCS have a laterality effect (seizures stats with one hemisphere). It is worth considering analysing ipsilateral vs. contralateral hemispheres (by flipping hemispheres) rather than left vs. right.

Response:

We apologized for the unclear clinical description of patients with epilepsy in our study. In our study, all patients with epilepsy were MRI-negative, which indicated that the patients with TLE have no focal parenchymal abnormalities, hippocampal atrophy, or abnormal blurring of the gray and white matter. Since none of the patients had received surgical treatment up to the time of data preparation for this work, we lacked histological evidence. We cannot strictly distinguish the seizures stats with one hemisphere, and we therefore did not explore the laterality effect in this work. We have added details on the TLE and GTCS as follows (Pages 22):

(Page 22) "Two types of patients had negative presentation on diagnostic MRI (i.e., no lesion such as cortical dysplasia, benign brain tumor or hippocampal sclerosis was detected), and none had received surgical treatment up to time of data preparation for this work.

Q3: Table 1 needs to include a lot of clinical information. For example.

- o Seizure onset age
- o Laterality of seizure
- o Seizure burden
- o Medication history
- o Specific EEG findings

Response:

Thank the reviewer for this valuable suggestion. We have added the relevant information as the next page:

Demographics	HC	TLE	GTCS	Comparisons	
	(n=108)	(n=75)	(n=79)	Statistical value	P value
Sex (males/females)	59/49	43/34	54/25	$\chi^2 = 0.03$ ($P = 0.87$)	$\chi^2 = 3.59$ ($P = 0.06$)
handedness (left/right)	0/108	0/75	0/79	$\chi^2 = 0$ ($P = 1$)	$\chi^2 = 0$ ($P = 1$)
Age (years)	23.72 ± 4.15	25.72 ± 8.64	24.78 ± 7.84	$U = 3889$ ($P = 0.64$)	$U = 3769$ ($P = 0.17$)
Duration of illness (months)	—	111.90 ± 89.25	87.27 ± 93.71	—	—
Seizure onset age (years)	—	16.51 ± 8.71	17.88 ± 7.65	—	—
Treatment (yes/no)	—	55/9 [#]	51/19 [#]	—	—
Seizure frequency (month)	—	6.06 ± 19.50	17.46 ± 80.88	—	—

Q4: Was head motion different between any of the sub-groups?

Response:

We apologize for the missing statements. The sub-groups were age-, sex-, and head motion (mean FD) matched.

Q5: What subtypes of Genetic Generalised Epilepsy did subjects with GTCS have?

Response:

In our study, participants were idiopathic generalized epilepsy with GTCS. We modified the information in the revision as follows (Page 22):

(Page 22) "For genetic generalized epilepsy with GTCS, the patients presented with a) typical seizure semiology of GTCS, including loss of consciousness during seizures without precursory symptoms of partial epilepsy and aura, and tonic extension of the limbs followed by a clonic phase of rhythmic jerking, b) generalized spike-and-wave discharges on EEG, and c) no other epilepsy associated etiology such as trauma, tumor, or intracranial infection."

Q6: The authors state that: "All patients had negative presentation on conventional (structural) magnetic resonance images (MRI), and a portion of patients had concordant positron emission tomography or magnetoencephalogram examination". Does this mean that GTCS and TLE were lesion-negative? This should be highlighted further in the manuscript because this isn't a more conventional hippocampal sclerosis group.

Response:

Yes, the TLE and GTCS were lesion-negative. Thanks for the valuable suggestions. We have extended the description of the clinical information of patients as follows (Page 22):

(Page 22) "Two types of patients had negative presentation on diagnostic MRI (i.e., no lesion such as cortical dysplasia, benign brain tumor or hippocampal sclerosis was detected), and none had received surgical treatment up to time of data preparation for this work."

Reviewer #3:

Q1: Authors study changes in macroscopic functional connectivity, as well as in parameters of the mesoscale mean field models in two types of epilepsy, as compared with a control group. The methodology seems solid and follows the state of the art, the size of the groups is also decent, but the results are less clear and better contextualizing them would improve the work. For example, is there any reason why the somatomotor regions would be mostly impacted? Similarly, from the beginning is not clear why the emphasize is on the gradient, instead of for example RSNs, or simply staying at nodes' level.

Response:

We greatly appreciate the insightful comments on our manuscript. We made modifications according to the suggestions and included point-by-point responses to each issue.

First, we cited some of the previous literature that found functional dynamic changes in somatomotor areas in epilepsy, and we have extended the discussion why the functional dynamics in somatomotor regions were impacted in epilepsy as follows (Page 17).

Second, with the Desikan–Killiany anatomical parcellation with 68 cortical regions, we extracted over 7,000 temporal features for each region. The gradient was obtained by identifying the main axes of variance in the temporal features matrix, which can be regarded as a second-order relationship. From the perspective of dimensionality reduction method, the gradient more focused on the relationship between brain regions with regard to their similarity of 5,000 shared temporal features. Resting-state network is more concerned with the time series correlation between regions, which can be regarded as a first-order relationship. As a result, the gradient and resting state networks capture different information about interregional relationships. We modified this information in the revision as follows (Page 6):

(Page 17) "The somatomotor cortices are integrated into a multimodal network associated with sensory, motor systems, and cognitive hub regions³⁵. By leveraging computational simulations that quantify patterns of information flow across the connectome, TLE showed altered communication dynamics between the somatomotor cortices and subcortical regions, and TLE also showed slower signal spreading time in high-order network with impairment in sensorimotor, executive, and memory functions³⁶. More specifically, GTCS has shown abnormal intrinsic brain activity in the primary network, as well as in internetwork communication frameworks. EEG–fMRI in patients with idiopathic generalized epilepsy demonstrated that increased connectivity with sensorimotor regions before the onset of discharges can trigger generalized spike wave discharges, involving network changes in the prefrontal cortex and precuneus³⁷. Together, the somatomotor cortices are implicated in two epileptic subtypes."

(Page 6) "From the perspective of dimensionality reduction method, gradient more focused on the relationship between brain regions regard to their similarity of 5,000

shared temporal features, which representing the fingerprints of regional intrinsic dynamics. We therefore emphasized the gradient of temporal features rather than temporal correlation of time-series.

Bibliography in the Response to Reviewers:

- 35 Sepulcre, J., Sabuncu, M. R., Yeo, T. B., Liu, H. & Johnson, K. A. Stepwise connectivity of the modal cortex reveals the multimodal organization of the human brain. *J Neurosci* 32, 10649-10661, doi:10.1523/JNEUROSCI.0759-12.2012 (2012).
- 36 Girardi-Schappo, M. et al. Altered communication dynamics reflect cognitive deficits in temporal lobe epilepsy. *Epilepsia* 62, 1022-1033, doi:10.1111/epi.16864 (2021).
- 37 Tangwiriyasakul, C. et al. Dynamic brain network states in human generalized spike-wave discharges. *Brain* 141, 2981-2994, doi:10.1093/brain/awy223 (2018).

Q2: Another weak point is the quite coarse parcellation that is used. At least for TLE, Virtual Epileptic Brain (VEP) atlas seems to offer better distinction between the regions of interest. The results would have been much strengthened if the analysis is confirmed in this, or in some other finer parcellation.

Response:

Thank the reviewer for this valuable suggestion. For analyses of massive temporal features extraction by *hctsa* toolbox and neural dynamics simulation in pMFM, finer parcellation means more computational and time consuming. Specifically, using the Desikan–Killiany anatomical parcellation with 68 cortical regions, total time used for the training processing of pMFM for one group is around one week.

We agree that VEP atlas offer better distinction between regions of interest for TLE. But for some reason, we re-analyzed our main findings using the Schaefer 200 parcel atlas. We will consider using the VEP in our further work. The details of the analysis are in the revision as follows (Pages 15 and 25):

(Page 15) “We replicated our results with a higher resolution parcellation with 200 cortical regions³³. Consistent with our main results, we found that the PC1 shows a ventromedial-lateral gradient and the PC2 captures the sensory-association gradient (Fig.S3A), and the RC increased from sensory-motor to association cortices and the I decreased from sensory-motor to association cortices (Fig.S3B). For the associations in multiscale brain function, we found negative correlation between the differences of PC1 and the changes of RC in TLE ($\rho = -0.22$, $P = 0.0015$) and GTCS ($\rho = -0.27$, $P < 0.001$) (Fig.S3C).”

(Page 25) “We additionally replicated the results with a higher resolution parcellation with Schaefer 200 cortical regions in controls analysis (Supplementary materials).

Fig.S3 With a higher resolution parcellation with Schaefer 200 cortical regions, spatial pattern of the PC1 and PC2 (A), and the RC and I (B). Network-wise differences in PC1 and PC2 were compared (B). Associations between the PC1 and RC in two epileptic subtypes (C).

Q3: In the introduction, the authors should mention the work by Courtiol et al. J neurosci 2020, and possibly other works that analyzed the rs fMRI in epileptic patients, such as Wirsich et al. Neuroimage 2016.

Response:

Thank the reviewer for this valuable suggestion. We have mentioned these work in the introduction as follows (Pages 3 and 4):

(Page 3) *“The Virtual Brain model has demonstrated that a combination of a global shift in the brain’s dynamic equilibrium and locally hyperexcitable network nodes provides a mechanistic explanation for the epileptic brain during interictal resting state⁹.*

(Page 4) *“The network communication analysis revealed that the enhanced structure-function correlation may be related to the smaller functional repertoire in TLE, while sparing the central core of the brain, which may represent a pathway that promotes the spread of seizures¹⁸.*

Bibliography in the Response to Reviewers:

- 9 Courtiol, J., Guye, M., Bartolomei, F., Petkoski, S. & Jirsa, V. K. Dynamical mechanisms of interictal resting-state functional connectivity in epilepsy. J Neurosci, doi:10.1523/JNEUROSCI.0905-19.2020 (2020).32 Girardi-Schappo, M. et al. Altered communication dynamics reflect cognitive deficits in temporal lobe epilepsy. Epilepsia 62, 1022-1033, doi:10.1111/epi.16864 (2021).
- 18 Wirsich, J. et al. Whole-brain analytic measures of network communication reveal increased structure-function correlation in right temporal lobe epilepsy. Neuroimage Clin 11, 707-718, doi:10.1016/j.nicl.2016.05.010 (2016).

Q4: instantaneous interactions are assumed between the brain regions, even though time-delays due to axonal propagation can be very important for the whole-brain dynamics, e.g. for synchronization. This needs to be justified and mentioned in the limitations.

Response:

Thank the reviewer for this valuable suggestion. We have clarified it in the Limitations as follows (Page 20):

(Page 20) *“First, even though time delays due to axonal propagation can be of crucial important for the study of the whole-brain dynamics, here we assumed instantaneous interactions between the brain regions and assumed that their impact to be encompassed in the neural masses, and we neglected them⁹.*

Bibliography in the Response to Reviewers:

- 9 Courtiol, J., Guye, M., Bartolomei, F., Petkoski, S. & Jirsa, V. K. Dynamical mechanisms of interictal resting-state functional connectivity in epilepsy. J Neurosci, doi:10.1523/JNEUROSCI.0905-19.2020 (2020).32 Girardi-Schappo, M. et al. Altered communication dynamics reflect cognitive deficits in

temporal lobe epilepsy. *Epilepsia* 62, 1022-1033, doi:10.1111/epi.16864 (2021).

Q5: In the Limitations it is worth mentioning about the possible limitations of the neuronal masses in general, which can be overcome by data-driven brain network models, such as one by Sip et al. *Sci Adv* 2023.

Response:

An inspiring piece of work. We have mentioned about the possible limitations of the neuronal masses in the Limitations as follows (Page 20):

(P20) "Second, the pMFM were derived through a series of major simplifications to arrive at simple, low-dimensional models of neural dynamics, and the results may depend on the exact model form. A data-driven approach that would allow us to avoid choosing a specific neural mass model and instead extract this model directly from the functional data⁴⁵. More important, this data-driven approach can estimate the link between dynamically relevant parameters and the measurable from a preexisting patient cohort, and then applied to a single subject, that might be more helpful for individualized patient estimation."

Bibliography in the Response to Reviewers:

45 Sip, V. et al. Characterization of regional differences in resting-state fMRI with a data-driven network model of brain dynamics. *Sci Adv* 9, eabq7547, doi:10.1126/sciadv.abq7547 (2023).

Q6: figures are not very sharp (especially bad are 2 and 3), vector graphics should be used if possible.

Response:

We have improved the quality of these figures in the revision. We upload the figure as a single file, attached to the end of the document.

Q7: l450 there seems to be a missing word in the sentence

Response:

We apologized for the misleading statements. We modified the sentence in the revision as follows (Page 27):

(P27) "The nonlinear stochastic differential equations of neural activity in each cortical region followed the same parameter settings as in the previous study²⁸."

Reviewers' comments:

Reviewer #1 (Remarks to the Author):

I reviewed this paper by Yang et al and unfortunately don't find their responses to be completely compelling. Primarily I remain concerned about the redundancy and circularity present in their analysis, particularly pertaining to the hctsa gradients. The authors claim that the time series (hctsa) gradients (from PCA) are different from the functional connectivity gradients (cf. Margulies). And while this is true in so far as the former is a projection of (lots of) univariate timeseries features while the latter is a projection of pairwise relationships between timeseries, they are still fundamentally both based on low dimensional representations of the same underlying data type (fMRI data). As such, the gradients of hctsa will likely correlate (and probably very strongly) with the gradients of FC. Thus, when the authors inform/constrain their biophysical model using the FC gradient (see Fig 1) only to then study how the parameters of that model relate to hctsa gradients (e.g. Fig 5), they are conducting highly circular analyses, and it remains unclear how even an SA preserving null overcomes this.

Additionally, even after their comments, it still remains unclear to me what the value of using hctsa is in the first place since the authors project the features to a low dimensional space. I suspect that authors would get more-or-less the same time series gradients (and thus the same results) by just looking at a couple of simple time series properties commonly adopted by the fMRI literature (e.g., bold amplitude, low frequency power etc.).

Reviewer #2 (Remarks to the Author):

The authors have addressed all my previous concerns. This is a good contribution to the literature.

Reviewer #3 (Remarks to the Author):

I'm happy with the authors' responses to my comments, and I recommend the article to be accepted for publication.

Response to Reviewers (COMMSBIO-23-2881B)

=====

Reviewer #1:

I reviewed this paper by Yang et al and unfortunately don't find their responses to be completely compelling. Primarily I remain concerned about the redundancy and circularity present in their analysis, particularly pertaining to the hctsa gradients. The authors claim that the time series (hctsa) gradients (from PCA) are different from the functional connectivity gradients (cf. Margulies). And while this is true in so far as the former is a projection of (lots of) univariate timeseries features while the latter is a projection of pairwise relationships between timeseries, they are still fundamentally both based on low dimensional representations of the same underlying data type (fMRI data). As such, the gradients of hctsa will likely correlate (and probably very strongly) with the gradients of FC. Thus, when the authors inform/constrain their biophysical model using the FC gradient (see Fig 1) only to then study how the parameters of that model relate to hctsa gradients (e.g. Fig 5), they are conducting highly circular analyses, and it remains unclear how even an SA preserving null overcomes this.

Response:

We appreciate your comments to improve the quality of the manuscripts.

We agree that both time series gradients and the functional connectivity (FC) gradients are fundamentally based on low dimensional representation of fMRI data. According to your suggestion, we checked the correlation between two gradients and found that PC1 (from *hctsa*) did not correlate with the FC gradient ($r = 0.018$, $p = 0.884$), but PC2 was negatively correlated with FC gradient ($r = -0.945$, $p < 0.001$) in health. These results were consistent with the work by Shafiei et al.

When analyzing how the parameters of the biophysical model relate to *hctsa* gradients, we focused on the spatial correlation of the group differences in both characterizations, rather than the primarily parameters and gradients in each group. An SA preserving did eliminate the circularity of the analysis; it only addressed the spatial autocorrelation.

Based on these suggestions, we modified statements in the revision as follows (Pages 13 and 20):

(Page 13) "Notably, we focused on the spatial correlation of the group differences in both characterizations, rather than the parameters and gradients in each group."

(Page 20) "While intrinsic dynamics gradient and functional connectivity gradient were fundamentally both based on low dimensional representations of the fMRI data. And we inform the biophysical model using the functional connectivity gradient (Figure 1) to then study how the model parameters of relate to intrinsic dynamics gradient (Figure 5). To avoid circular analysis, we focused on the spatial correlation of the group differences in intrinsic dynamics gradient and microcircuit parameters, rather than the parameters and gradient itself in each group."

Bibliography in the Response to Reviewers:

Shafiei, G. et al. Topographic gradients of intrinsic dynamics across neocortex. *Elife* 9, doi:10.7554/eLife.62116 (2020).

Additionally, even after their comments, it still remains unclear to me what the value of using *hctsa* is in the first place since the authors project the features to a low dimensional space. I suspect that authors would get more-or-less the same time series gradients (and thus the same results) by just looking at a couple of simple time series properties commonly adopted by the fMRI literature (e.g., bold amplitude, low frequency power etc.).

Response:

Thanks for this valuable suggestion. Yes, the time series gradients could derive from a couple of simple time series properties commonly adopted by the fMRI literature. These conventional computational analyses are based on specific, manually selected time-series features. Yet the time-series analysis literature is vast and interdisciplinary, we therefore chose the *hctsa* toolbox to get a summary characterization. But some of these features are highly similar, projecting to low dimensional space could help capture top most explained components. The exploration methods for epileptic brain diseases should be diversified, and our analysis can provide a complementary analysis in the research. We are willing to make other exploration in the future study.

We modified statements in the revision as follows (Page 18):

(Page 18) "Rather than selecting conventional or specific time-series properties, the advantage of using hctsa is obtaining massive temporal features to get a summary characterization."

REVIEWERS' COMMENTS:

Reviewer #1 (Remarks to the Author):

The authors have addressed my comments.